# The Effect of B_4_C Powder on Properties of the WAAM 2319 Al Alloy

**DOI:** 10.3390/ma16010436

**Published:** 2023-01-03

**Authors:** Xueping Song, Jinke Niu, Jiankang Huang, Ding Fan, Shurong Yu, Yuanjun Ma, Xiaoquan Yu

**Affiliations:** 1State Key Laboratory of Advanced Processing and Recycling of Non-ferrous Metals, Lanzhou University of Technology, Lanzhou 730050, China; 2Mechanical Engineering College, Lanzhou Petrochemical University of Vocational Technology, Lanzhou 730060, China; 3Materials Science and Engineering College, Lanzhou University of Technology, Lanzhou 730050, China; 4Mechanical and Electrical College, Lanzhou University of Technology, Lanzhou 730050, China; 5Research Institute of Zhejiang University–Taizhou, Zhejiang University, Taizhou 318000, China

**Keywords:** 2319 aluminum alloy, microstructure, B_4_C powder, WAAM, mechanical properties

## Abstract

With ER2319 and B_4_C powder as feedstocks and additives, respectively, a wire arc additive manufacturing (WAAM) system based on double-pulse melting electrode inert gas shielded welding (DP-MIG) was used to fabricate single-pass multilayer 2319 aluminum alloy. The results showed that, compared with additive manufacturing component without B_4_C, the addition of which can effectively reduce the grain size (from 43 μm to 25 μm) of the tissue in the deposited layer area and improve its mechanical properties (from 231 MPa to 286 MPa). Meanwhile, the mechanical properties are better in the transverse than in the longitudinal direction. Moreover, the strengthening mechanism of B_4_C on the mechanical properties of aluminum alloy additive manufacturing mainly includes dispersion strengthening from fine and uniform B_4_C granular reinforcing phases and fine grain strengthening from the grain refinement of B_4_C. These findings shed light on the B_4_C induced grain refinement mechanism and improvement of WAAM 2319 Al alloy.

## 1. Introduction

In recent years, with the rapid development in additive manufacturing (AM) of metal components, layer-by-layer component manufactured by AM has received wide application [1]. Indeed, 2319 aluminum alloy, an Al-Cu-Mn-Mg-Ag alloy strengthened by Al_2_Cu precipitates, is light, corrosion-resistant and anti-fatigue, and has been widely used in the aerospace, military, and automotive industries, among others. The introduction of heterogeneous particles can effectively improve the overall performance of 2319 aluminum alloy with refinery grain size and better mechanical properties. Studies have shown that appropriate addition of ceramic or metal powders can improve the microstructure and mechanical properties of alloys. The ceramic powder in the aluminum alloy acts as a pegging dislocation and sub-grain boundary in the aluminum matrix, hindering dislocation and grain boundary migration. It also prevents recrystallisation grain nucleation and growth, and increases the recrystallisation temperature. According to some researchers, because of poor printability and high susceptibility to cracking of 2319 aluminum alloys, inoculants are added to improve the printability of deformed aluminum alloys [2]. Many effective inoculants such as TiC [3], TiB + TiC [4], ZrH_2_ [5], SiO_2_ [6] and Er [7] have been proposed for application in crack-sensitive alloy, in which the introduction of nanoparticles as nucleating agents in fissile alloys induced a large number of equiaxed crystals with strong strain regulation, thus effectively inhibiting thermal cracking during solidification.

Boron carbide (B_4_C) with excellent physical and mechanical properties is characterized by high hardness and good resistance to chemical agents, which makes it widely used in various fields such as military, engineering, nuclear energy and high temperature thermoelectric conversion [8]. In the family of deformed aluminum alloys, the 2××× series aluminum alloys are good candidates for AM machining due to their good weldability [9,10,11,12,13,14,15,16,17,18]. In general, the deposited additive manufactured aluminum alloys are inferior to conventionally processed ones in mechanical properties [19]. K.S. Sridhar Raja et al. [20] and Bhargavi Rebba et al. [21] explored the microstructure and mechanical properties of the fabricated AMCs through the combination of aluminum 2024 alloy reinforced and B_4_C with weight% 0%, 1%, 2%, 3%, 4% and 5%. To study the improvement in mechanical properties after the reinforcement particles were imparted to the composites, T. Raviteja et al. [22] produced composites (Al-Si_12_Cu/B_4_C) by reinforcing B_4_C (33 μm) with varying wt% of 2, 4, 6, 8 and 10 through stir casting process. A. Baradeswaranet et al. [23] investigated the effect of B_4_C (with 5, 10, 15 and 20 vol% B_4_C particles with particle size ranging from 16 μm to 20 μm) on the mechanical and tribological behavior of 7075 Al composites.

Grain Structure of AM parts can be controlled through AM process and post heat-treatments [24]. However, there are few reports on the effect of boron carbide (B_4_C) on additively manufactured parts from aluminum alloys. Therefore, inspired by the above research, we innovated the use of boron carbide particles as heterogeneous nucleating agents to add to the additive manufacturing molten pool, in order to improve the nucleation rate and limit the grain size, so as to refine the grain effect. In this paper, 2319 aluminum alloy additive manufacturing is achieved using the DP-MIG method, and good shape aluminum alloy parts are thus obtained by varying the melting rate. The microstructure of aluminum alloy additive parts under different process parameters was analyzed to investigate the influence of different melting rates and the addition of appropriate amount of B_4_C powder on the macroscopic morphology, composition distribution, microstructure and phase composition of additive manufacturing specimens. Finally, the aluminum alloy samples were subjected to tensile tests in the vertical and transverse directions, respectively.

## 2. Experimental Details

The feedstock is ER2319 aluminum alloy wire with a diameter of 1.2 mm. The substrate used in the experiment is 1060 aluminum alloy with a size of 150 mm × 100 mm × 5 mm. Table 1 lists the main chemical composition of the wire and the substrate. The WAAM system based on DP-MIG used in the experiment is shown in the Figure 1. The experiment analyzes the aluminum alloy additive manufacturing processes at different melting rates, with the addition of B_4_C powder aluminum alloy additive manufacturing processes at a better processing parameters. DP-MIG torch is at 90° to base material and the powder is blown out by the shield gas flow in the powder feeder at 20 L/min. The powder injection rate is 25 g/min. Moreover, the thermocouple is fixed to the aluminum substrate using a welding machine before the test. More points were measured during the experiment and averaged for accuracy.

Considering the forming quality is largely dependent on the melting speed in the double-pulse MIG aluminum alloy additive manufacturing process [25], a variable is introduced in a series of relevant experiments as shown in Table 2. The crystal and phase structural information of the samples was characterized using an X-ray diffractometer (XRD, D8ADVANCE) with Cu Ka radiation. Scanning electron microscope (SEM) was employed for microstructural characterization. The size of precipitates was analyzed based on SEM images. The element distribution was scanned by electron probe micro analysis (EPMA, JXA-8530F).

The measurement of the tensile properties of thin-walled aluminum alloy deposited layers in the fusion direction (transverse) and thin-walled deposited layers in the vertical direction (longitudinal) are shown in Figure 2a. Dog-bone-shaped specimens (whose dimensions are shown in the Figure 2b) were produced for tensile tests using an electro discharge machine. The uniaxial tensile tests were conducted on an Instron 8802 tension machine at room temperature with a strain rate of 1.0 × 10^3^ s^−1^. For accuracy, three tensile specimens were prepared simultaneously in the same direction for the experiment. Microhardness measurements were performed for sample by a QATM Q10A+ automatic hardness tester with parameters of 10 g and 10 s.

## 3. Results

### 3.1. Formation of Deposited Layer

Figure 3 shows the shape of the aluminum alloy add-on at different melting rates at a fixed low frequency of 3 Hz. With the deposition rates at 120, 130, 140 and 150 mm/min, the width of the aluminum addendum decreases with increasing melt rate. When the deposition rate is 150 mm/min, the heat input is small. Although a good metallurgical bond is achieved between the layers, some aluminum alloys are not completely melted, and a layer of the formed area appears inclined and uneven in thickness. In addition, when the deposition rates is at 140 mm/min, the bottom layer of the sidewall of the deposition piece has bulges that cannot be fused, but the surface of the structural piece is smooth and flat. Meanwhile, collapse was observed at the end of the aluminum alloy additive manufacturing specimen. The main reason is the interaction of arc pressure and molten droplets which makes the molten metal move towards the tail of the molten pool, resulting in the accumulation of a large amount of liquid metal at the end of the molten pool, which eventually solidifies and forms. However, the deeper the deposition, the more severe the collapse is [26,27].

### 3.2. Microstructure Morphology

When the deposition current is 80A, the microstructure of WAAM 2319 aluminum alloy at different deposition rates is shown in Figure 4. The microstructure of aluminum alloy without B_4_C filling is shown in Figure 4a–d. It can be seen that the microstructure is not uniform, with some large grains and small sized grains. When B_4_C is filled, the microstructure of the aluminum alloy is shown in Figure 4e, in which the microstructure is significantly refined, basically consisting of fine equiaxed grains, with individual large second phase and no obvious oxide accumulation.

As seen in Figure 5, the X-ray diffraction analysis measures the angular range from 10 to 90°. Obvious α-Al and θ-Al_2_Cu phases were observed in all the WAAM aluminum alloy. The B_4_C phase was found in the 140 mm/min-B_4_C in which B_4_C powder was added, as shown in Figure 5d. It can be seen that no phase transition occurs in the WAAM 2319 aluminum alloy. In addition, during the preparation of WAAM aluminum alloy, oxygen elements were inevitably let in, and the aluminum alloy particles underwent oxidation reaction, resulting in the formation of Al_2_O_3_ and other substances in the alloy. These substances show no obvious diffraction peaks in the XRD diffraction pattern, which may be due to the small amount of these substances, the weak strength of their diffraction peaks, or the overlap of diffraction peaks with those of other substances.

Figure 6 shows the macroscopic morphology and energy spectrum analysis of B_4_C powders which can be seen as rectangular or even short rods with an uneven surface and a powder diameter of approximately 50~70 μm. In addition, without any remaining impurities, B_4_C powder has a good surface finish and integrity and thus ensures the quality of aluminum alloy forming.

The microstructure and composition of the deposited parts were observed at a melting rate of 140 mm/min. Figure 7 shows the EDS surface scan analysis without B_4_C powder, in which the microstructure of 2319 aluminum alloy is found to consist mainly of Al and Cu elements, and the SEM backscatter pattern shows a large amount of second phase is attached to the aluminum matrix in addition to a honeycomb network of eutectic tissue distributed along the grain boundaries. Four points in the microstructure were selected for EDS analysis, with point A selected at the Al alloy matrix, points B and C both at the grain boundaries and point D at the honeycomb network of eutectic tissue. The results of the EDS analysis of the aluminum alloy specimens in the deposited state is shown in Table 3. According to XRD analysis, the microstructure of 2319 aluminum alloy consists mainly of α-Al and θ-Al2Cu phases [28]. During the melting process, most of the Cu elements are solidly dissolved into the α-Al matrix and precipitate to form a θ-Al_2_Cu reinforced phase during melting and cooling. In addition, the results at point A indicate that there is some Cu in the aluminum matrix besides the α-Al. Comparing the results of the EDS compositional analysis at the four points locations in Figure 7a, it is found that point C has the highest Cu content, while point B has a lower Cu content, implying that the Cu elements in the aluminum alloy deposit layer are not uniformly distributed. The composition ratio of the second phase at point D is close to that of the Al-Cu eutectic, which identifies the honeycomb organization in the grain boundary as α-Al and θ-Al_2_Cu eutectic. As can be seen from Figure 7, when the precipitated phase is generated, a large amount of elemental Cu is present in the form of the precipitated phase and only a small proportion of elemental Cu can be observed in the matrix.

Figure 8 shows the microstructure and EDS composition of the aluminum alloy deposited at a deposition rate of 140 mm/min with the addition of B_4_C powder. A map scan analysis of the B_4_C powder within the grain boundaries shows that the B_4_C powder is composed of element C and element B. The SEM backscatter pattern shows a large amount of second phase and B_4_C powder distribute in the aluminum matrix and a small amount of honeycomb net-like eutectic tissue around the grain boundaries. Four points in the microstructure are selected for EDS analysis, that is, point A is selected on the Al matrix, point B on the honeycomb mesh, point C on the B_4_C powder and point D on the granular second phase. Table 4 shows the results of EDS analysis of aluminum alloy specimens in the deposited state with the addition of B_4_C powder. The composition of point A, almost entirely Al, is close to that of the original specimen, suggesting that the likely composition of the physical phase at point A is α-Al. The distribution of elements at point B is similar to that without the addition of powder in terms of both α-Al and θ-Al_2_Cu eutectic structures. Point C contains only element C and element B, indicating that point C is a B_4_C powder. Additionally, the second phase morphology is essentially granular, with no bright, fine, needle-like second phases found. As can be seen from Figure 8, the morphology of B_4_C powder is mainly in the form of plates or granules, uniformly distributed within the grain boundaries and tightly bonded with the α-Al matrix.

### 3.3. Mechanical Properties

The microhardness values indirectly reflect the mechanical properties of the aluminum alloy in the deposited state. Figure 9 is the microhardness of aluminum alloys in the deposited state with different melting rates and the addition of B_4_C powder. The microhardness value increases with the increase in the melting rate. To be specific, the microhardness of the deposited aluminum alloy reaches 91.6 HV after the addition of B_4_C powder and without B_4_C powder it is 81.4 HV, representing an increase of 12.5%. With the fill of B_4_C particle, the increase in aluminum alloy microhardness can be attributed to the following two aspects: B_4_C particles are hard materials with hardness up to 60 GPa. Hard B_4_C particles are evenly distributed in the matrix aluminum alloy, which improves the overall hardness of aluminum alloy. On the other hand, the interface between B_4_C particles and matrix formed a good interface bonding, B_4_C particles hindered the dislocation movement in the matrix aluminum alloy, causing a lot of dislocation plugging around the particles, and improvement in the hardness of the composites.

To study the mechanical properties of 2319 aluminum alloy additions in different orientations in the deposited state on the basis of a melting current of 80 A, the tensile specimen is cut from a single wall, and divided into transverse and vertical directions. In each direction, three tensile specimens are selected for tensile testing and averaged. Figure 10a shows the stress–strain curve of the aluminum alloy tensile specimen in the transverse direction and Figure 10b shows the stress–strain curve of the aluminum alloy tensile specimen in the longitudinal direction. The stress–strain curve shows that the tensile strength and elongation of the deposited aluminum alloy with B_4_C powder are much higher than those without B_4_C powder.

Table 5 shows the results of tensile specimens of deposited aluminum alloys in different orientations, whose mechanical properties are anisotropic in different directions. Overall, the mechanical properties are best in the transverse direction and are much higher than those in the longitudinal direction. When the deposition rates are at 120 mm/min, the deposited aluminum alloy has a tensile strength of 179.60 MPa in the transverse direction and an elongation of 5.89%. The tensile strength increases to 200.31 MPa and the elongation increases to 6.94% when the deposition rates increase to 130 mm/min. When the deposition rate is kept at 140 mm/min, the tensile strength is 285.84 MPa with the addition of B_4_C powder, representing an increase of 23.98%, and the elongation changes from 7.46% to 10.24%, an increase of 37.27%. In the longitudinal direction, the tensile strength is 270.9 MPa with the addition of B_4_C powder, an increase of 26.44%, and the elongation changes from 4.4% to 6.29%, an increase of 42.95%. The tensile strength and elongation in the longitudinal direction are much lower than in the transverse direction due to the weak bond between the deposited layers prone to defects during the deposition. As a result, the mechanical properties of the deposited aluminum alloy are obviously improved by adding B_4_C powder.

## 4. Discussion

This paper analyses the differences in microstructure and phase composition distribution at different deposition rates. The aluminum alloy deposition samples have better performance as regards forming at a deposition rate of 140 mm/min. It also discusses the difference in mechanical properties and how to improve the microstructure and mechanical properties of the aluminum alloy additive parts with addition of B_4_C powder.

### 4.1. Effect of Melting Rate on Forming Accuracy

Figure 11 is the layer width of specimen at different melting rates. As shown in Figure 11a, the width of the deposit layer varies with the increase in the number of layers. The width of the first five layers varies greatly, which increases gradually with the number of layers. When the number of layers is higher than six, the width of the layer tends to be gradually stable. As shown in Figure 11b, the stability zone in width height is in inverse proportion to melting rates. When the melting rate is reduced from 150 mm/min to 140 mm/min, the stability zone width increases from 10.64 mm to 11.78 mm (reduce 10.71%), and the stability zone height increases from 2.5 mm to 2.78 mm (increase 11.2%). It can be seen that the deposition rate has an obvious effect on the sample forming.

With the deposition rates at 120, 130, 140, 150 mm/min, and the deposition current at 80A, the sidewall morphology and roughness of aluminum alloy deposited parts are shown in Figure 12. When the deposition rates are slowed to 140 mm/min, the side wall roughness reduces to a minimum of 0.63. At this point, good form quality is achieved with wavy boundary between the layers and uniform thickness. Moreover, the side walls are bright and neat.

### 4.2. Effect of B_4_C Powder on Microstructure

2319 aluminum alloy is a precipitation-reinforced aluminum alloy, and the state of the grain organization changes drastically during the melting process after thermal cycling. Columnar and equiaxed crystals dominate the microstructure of the aluminum alloy at different melting rates. With the increase in the deposition rate, the grain size of the aluminum alloy deposited samples tends to decrease, as shown in Figure 13. When the deposition rates are 140 mm/min, the microstructure dominated by fine equiaxed crystals is uniformly distributed in the grain boundaries after addition of B_4_C powder; thus, there is a significant reduction in grain size in the microstructure, which proves that powders have an effect on the microstructure of 2319 aluminum alloy. The addition of B_4_C powder results in a significant refinement of grain size, due to the fact that the B_4_C powder can promote the heterogeneous nucleation of grains, the nucleation mass in the crystal and the nucleation rate. The significant increase in the number of grains per unit area, in turn, leads to grain refinement from 43 to 25 μm. Moreover, it increases the resistance to grain boundary migration resulting from the good stability of B_4_C powder when deposited on aluminum alloys and its ability of preventing dislocation at grain boundaries and within the grain during the melting process of the aluminum alloy, so that the grain boundary structure of aluminum alloy can be refined significantly.

### 4.3. Effect of B_4_C Powder on Mechanical Properties

SEM scanning shows the fractures of aluminum alloy-deposited samples in different orientations of tensile specimens. Figure 14 is the fracture profile of a deposited aluminum alloy tensile specimen in the transverse direction, where in the upper right corner an enlarged view of the corresponding fracture profile is displayed. As can be seen from the diagram, the transverse direction of the fracture contains a large number of tough nests whose center with addition of B_4_C powder contains second phase particles. The macroscopic fracture morphology shows that the pores in the fracture of the added B_4_C powder are significantly reduced, and the tough nests are uniformly small and densely distributed. It can be seen that the plastic deformation of B_4_C particles is not coordinated with that of the matrix aluminum alloy. The plastic deformation of aluminum alloy is large, while the plastic deformation of B_4_C particles is small. The addition of B_4_C particles has a strengthening effect on aluminum alloy. Figure 15 is the fracture profile of a deposited aluminum alloy tensile specimen in the longitudinal direction, where an enlarged view of the corresponding fracture profile is in the top right corner. The macroscopic morphology shows a clear necking phenomenon which suggests that interlayer stomatal aggregation and microstructural homogeneity are the main causes of anisotropy. Figure 15 shows that the large number of interlaminar-bond concentrated pores in the fracture of the longitudinal specimens while the fracture is at the interlaminar bond. Areas of pore accumulation in the longitudinal direction can form stress concentrations during stretching, creating a source of fracture. The effect of pores on tensile specimens in the transverse direction is less than that in the longitudinal direction, thus causing anisotropy in tensile properties in different directions. With the deposition rate at 140 mm/min, the fracture profile in the longitudinal direction without the addition of powder shows a small number of distinct deconfined surfaces. The enlarged view shows the laminar and fluvial nature of the fractured section, and some intergranular cracks are found near the dimples, both of which can act as fracture sources and play a detrimental role in the stretching process. When the deposition rate is 140 mm/min and B_4_C powder is added, there is neither deconstructed surface in the tensile fracture, nor large pore accumulation. Meanwhile, there exists broken second phase particles in the tough nest and a large number of tear ribs. In the longitudinal direction, the tensile specimens with different parameters fracture in a similar way and in a mixed fracture mode as well.

The presence of B_4_C particles for improving the mechanical properties of WAAM 2319 aluminum alloy can be attributed to the following aspects:

First of all, B_4_C particle plays a nucleating agent role in crystallization process, which can produce heterogeneous nucleation greatly. The grain size decreases and the grain boundary area increases after the addition of B_4_C powder. Thus, the fine grain strengthening effect is produced. The effect of grain refinement on strength can be described by the Hall–Petch formula σ_s_ = σ_0_ + Kd^−1/2^ [29], where σ_0_ represents the resistance to deformation within a crystal, K represents the degree of influence of grain boundary on strength, and d is the average diameter of each grain. After grain refinement, the ratio of grain boundary increases, and the effect of grain boundary hindering dislocation slip is strengthened. The plugging of dislocation near grain boundary leads to the increase in yield strength and tensile strength of aluminum alloy with grain refinement.

Secondly, the thermal expansion coefficient of B_4_C particles is different from that of aluminum alloy. In the process of preparation, the material constantly goes through the heating and cooling process, and dislocations are generated in the matrix aluminum alloy around the hard B_4_C particles. B_4_C particles play the role of nailing and blocking the dislocations, and improve the strength of the material. Moreover, the existence of B_4_C particles in the aluminum alloy makes the load transfer from the matrix aluminum alloy to the B_4_C particles in the process of load-bearing, which is conducive to the B_4_C particles bearing the transfer load, thus optimizing the mechanical properties of the material. In other words, there are fine and uniform granular reinforcing phases dispersed within the grain, resulting in a dispersion-strengthening effect. Reinforcing particles serve as a nailing effect on the dislocation, further hindering the dislocation movement and producing a strengthening effect.

Thirdly, B_4_C particles can effectively remove the oxygen element in the molten pool, which can reduce the aluminum oxide formed on the surface of the cladding layer in WAAM process. Thus, the melting effect between layers is better than that without B_4_C. It is one of the reasons for the improvement of its mechanical properties. In addition, the addition of B_4_C particles increases the fluidity of liquid metal in the molten pool, so as to improve the shape quality of deposition.

## 5. Conclusions

Based on the DP-MIG arc additive manufacturing technology, this paper uses ER2319 aluminum alloy welding wire as feedstock to conduct a single-pass thin-walled specimen forming test research. It investigates the surface morphology, deposited state organization and properties of thin-walled specimens of 2319 aluminum alloy at different deposition rates after the addition of B_4_C powders, whose role in double-pulse MIG arc additive manufacturing is also analyzed by exploring the microstructure, property defense and comparative second-phase changes in the deposited aluminum alloy. The main conclusions are as follows:(1)As the melting rate increases, the heat input gradually decreases, and the deposited aluminum alloy is formed with the highest accuracy at a deposition rate of 140 mm/min. The microstructure consists mainly of columnar crystals with only a small number of equiaxial crystals when the melting rate is small, and mainly of fine equiaxial crystals when the melting rate is large.(2)At a deposition rate of 140 mm/min and the addition of uniformly fine B4C powder, the B4C powder appeared in the microstructure of the deposited aluminum alloy. The grain size is reduced, and the microhardness and mechanical properties of the de-posited aluminum alloy are significantly improved, and there is anisotropy between the mechanical properties in the transverse and longitudinal directions, with better mechanical properties in the transverse direction than in the longitudinal direction.(3)B_4_C powder makes the grain size of the microstructure of the deposited layer smaller. As the grain size on the deposited layer becomes smaller, the mechanical properties of the deposited layer are obviously raised. Compared with the manufacturing without B_4_C, the microhardness, horizontal tensile strength, horizontal elongation, vertical tensile strength and vertical elongation of the deposited alloy are increased by 12.53%, 23.98%, 37.27%, 26.44% and 42.95% respectively.(4)The fracture mode is a mixture of destructive fracture and ductile fracture for different parameters, and the strengthening mechanism of B_4_C powder on the mechanical properties of aluminum alloy additive manufacturing is also revealed to be conducive to enhancing fine grain and precipitation phase.

## Figures and Tables

**Figure 1 materials-16-00436-f001:**
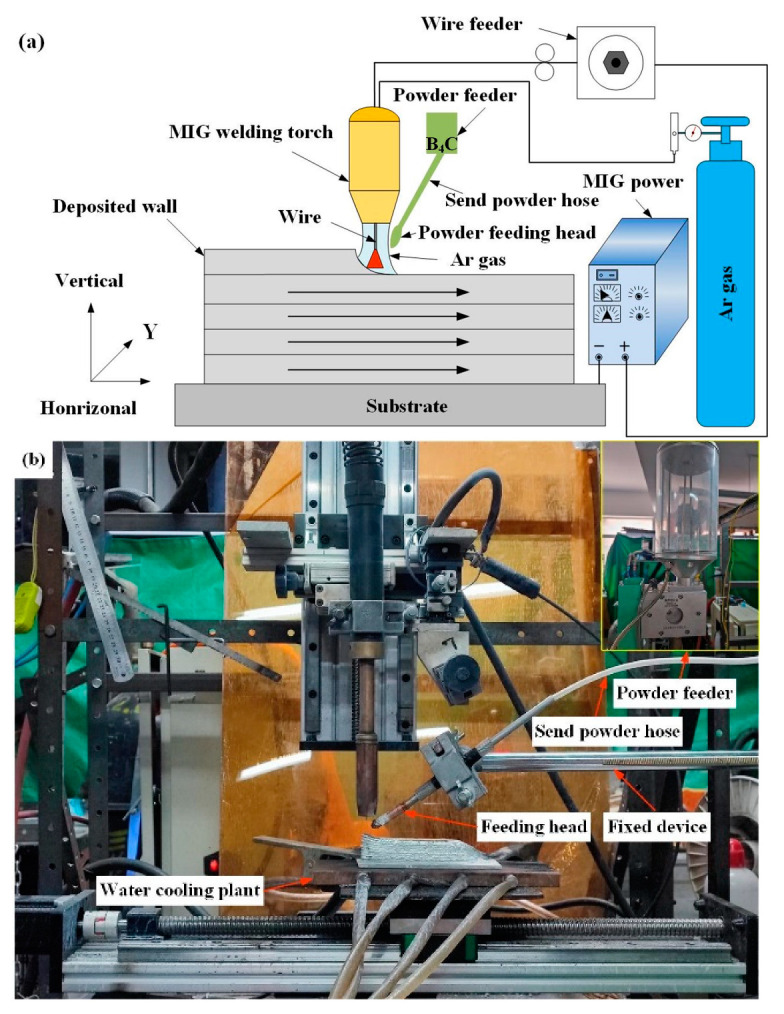
DP-MIG aluminum alloy powder feeding additive manufacturing. (**a**) schematic diagram of the DP-MIG-WAAM process, (**b**) structure of experimental equipment.

**Figure 2 materials-16-00436-f002:**
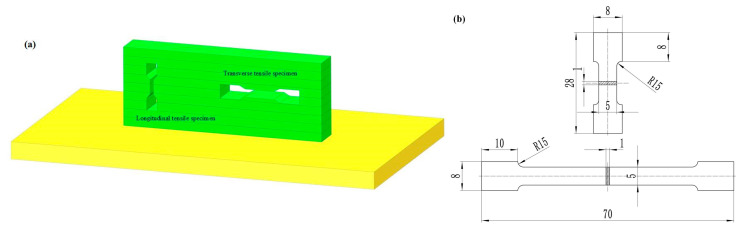
Specimen preparation (**a**) schematic diagram of specimen preparation, (**b**) specimen preparation dimensions.

**Figure 3 materials-16-00436-f003:**
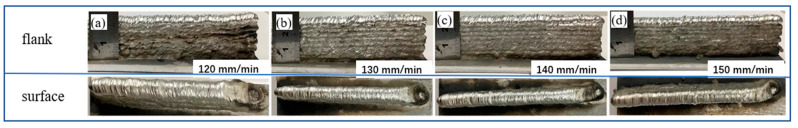
Forming of aluminum alloy additive manufacturing under different melting rates. (**a**) 120 mm/min, (**b**) 130 mm/min, (**c**) 140 mm/min, (**d**) 150 mm/min.

**Figure 4 materials-16-00436-f004:**
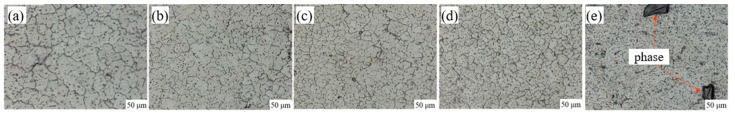
Microstructure of B_4_C particle with different deposition rates. (**a**) 120 mm/min, (**b**) 130 mm/min, (**c**) 140 mm/min, (**d**) 150 mm/min, (**e**) 140 mm/min-B_4_C.

**Figure 5 materials-16-00436-f005:**
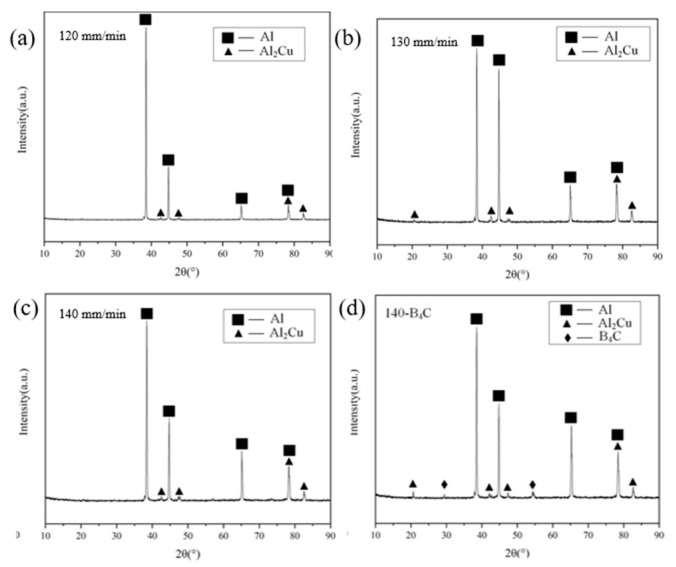
XRD analysis of different deposition rates (**a**) 120 mm/min, (**b**) 130 mm/min, (**c**) 140 mm/min, (**d**) 140 mm/min-B_4_C.

**Figure 6 materials-16-00436-f006:**
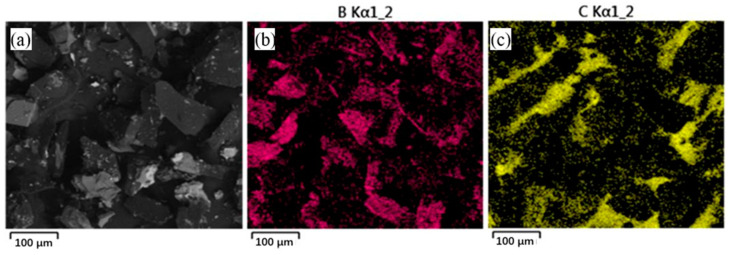
Macroscopic morphology and energy spectrum analysis of B_4_C powders. (**a**) 50 μm (**b**) 60 μm (**c**) 70 μm.

**Figure 7 materials-16-00436-f007:**
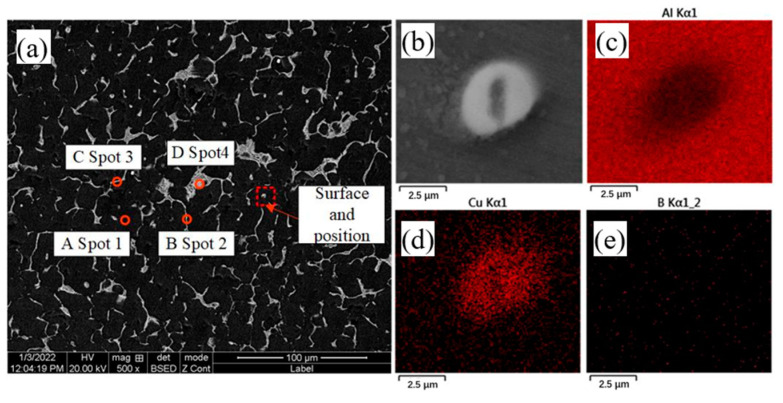
Microstructure and EDS composition analysis of B_4_C-free powdered aluminum alloy deposited parts (**a**) SEM analysis, (**b**–**e**) EDS analysis.

**Figure 8 materials-16-00436-f008:**
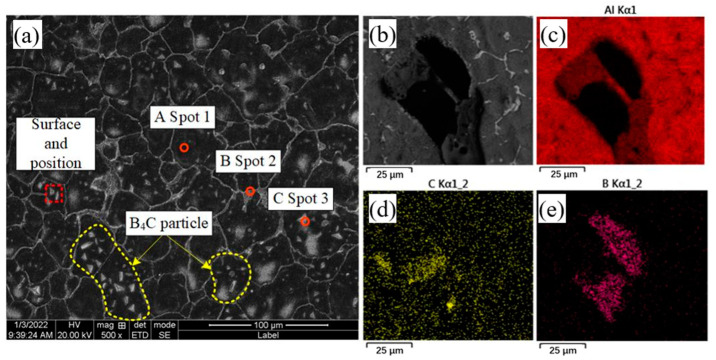
Microstructure and EDS composition analysis of deposited aluminum alloy parts with B_4_C powder. (**a**) SEM analysis, (**b**–**e**) EDS analysis.

**Figure 9 materials-16-00436-f009:**
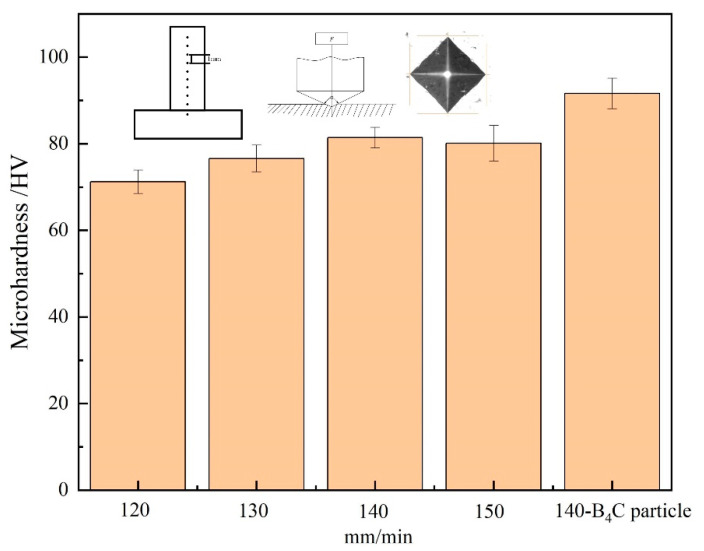
Microhardness values at different deposition rates with B_4_C powder.

**Figure 10 materials-16-00436-f010:**
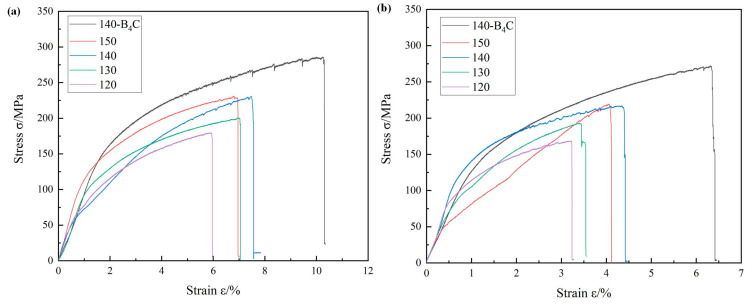
Stress–strain curves for aluminum alloys in the deposited state (**a**) transverse direction, (**b**) longitudinal direction.

**Figure 11 materials-16-00436-f011:**
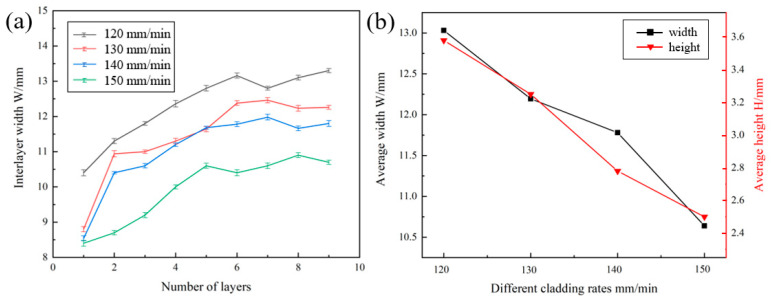
Variation of specimen layer width at different melting rates (**a**) number of specimen layers and width between layers, (**b**) deposition rate and average width and height.

**Figure 12 materials-16-00436-f012:**
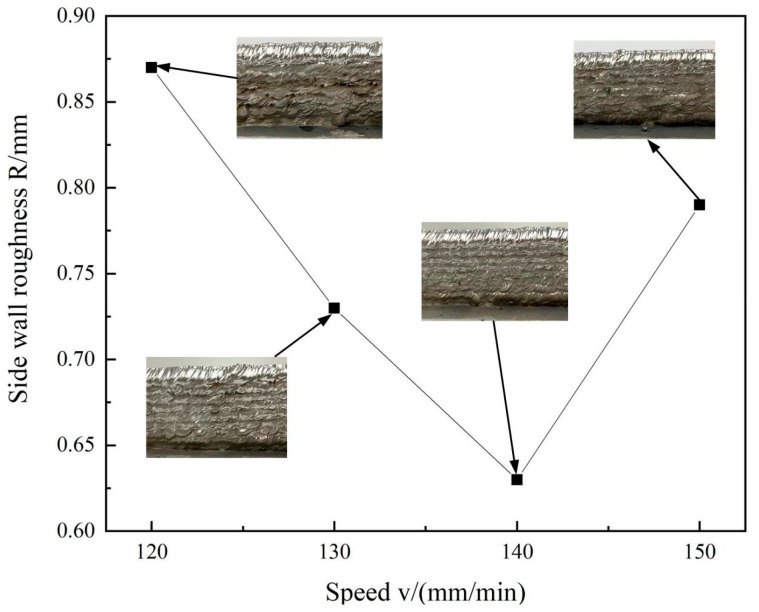
Sidewall morphology and its roughness at different deposition rates.

**Figure 13 materials-16-00436-f013:**
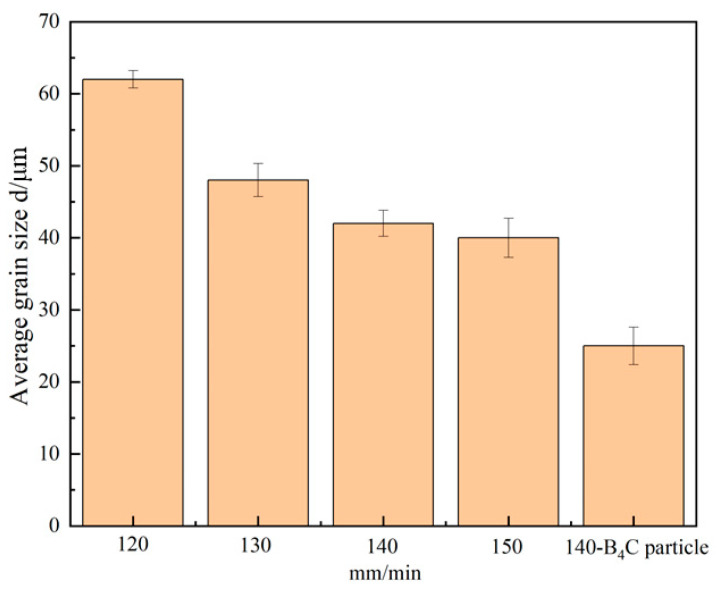
Effect of different deposition rates and addition of B_4_C particle on grain size.

**Figure 14 materials-16-00436-f014:**
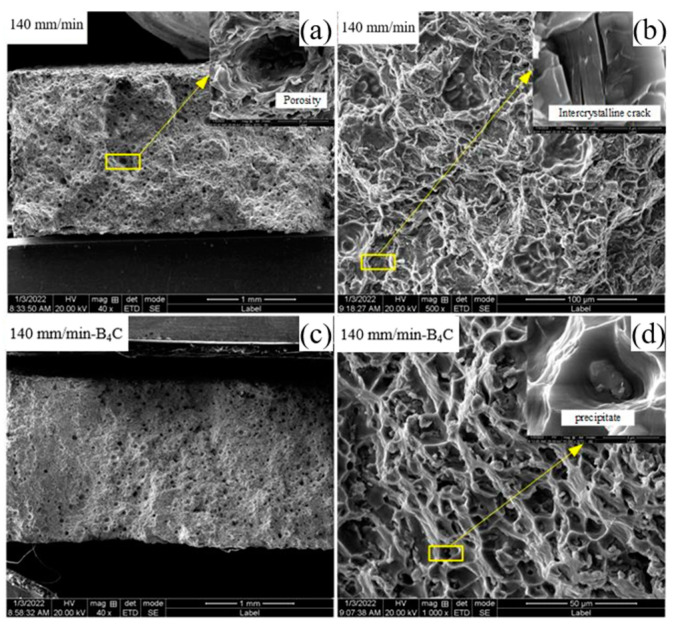
Fracture profile of deposited aluminum alloy tensile specimens in transverse direction (**a**,**b**) 140 mm/min, (**c**,**d**) 140 mm/min-B_4_C.

**Figure 15 materials-16-00436-f015:**
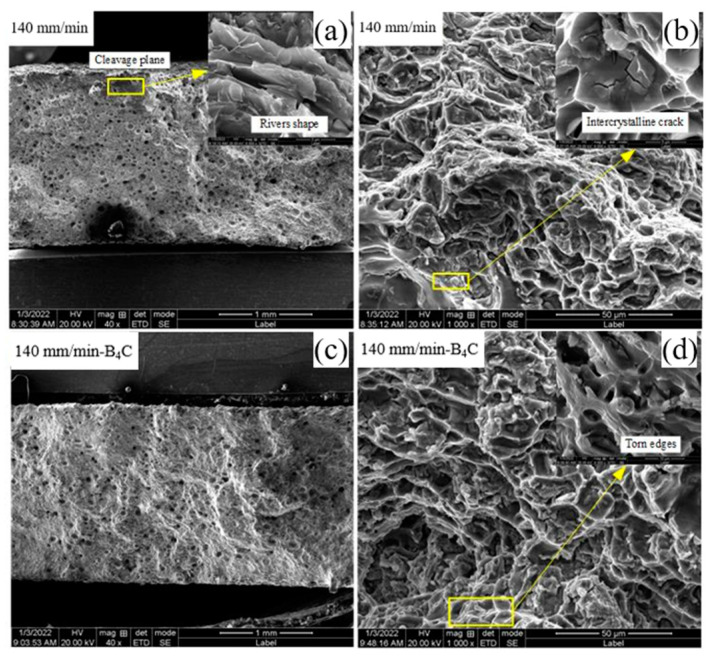
Fracture profile of deposited aluminum alloy tensile specimens in longitudinal direction. (**a**,**b**) 140 mm/min, (**c**,**d**) 140 mm/min-B_4_C.

**Table 1 materials-16-00436-t001:** Chemical compositions of 1060 pure aluminum substrate and ER2319 wire (wt.%).

Materials	Cu	Mg	Mn	Zr	Ti	V	Zn	Fe	Si	Al
2319	6.30	≤0.02	0.20	0.18	0.10	0.10	≤0.10	≤0.30	≤0.20	Del.
1060	0.05	≤0.03	0.03	0.18	0.03	0.05	≤0.10	≤0.35	≤0.25	Del.

**Table 2 materials-16-00436-t002:** Experiment parameters of DP-MIG wire welding additive manufacturing of 2319 aluminum alloy.

Number	Melting Current (A)	Duty Cycle (%)	Pulse Frequency (Hz)	Melting Speed (mm/min)	Shield Gas Flow Rate (L/min)	Powder Injection Rate (g/min)
1	80	50	3	120	0	0
2	80	50	3	130	0	0
3	80	50	3	140	0	0
4	80	50	3	150	0	0
5	80	50	3	140	20	25

**Table 3 materials-16-00436-t003:** The EDS analysis results for each point in Figure 7.

Spot	Al	Cu	Possible Material Phase Composition
1	98.26	1.74	α-Al
2	80.21	19.79	α-Al and θ-Al_2_Cu
3	59.65	40.35	α-Al and θ-Al_2_Cu
4	63.35	36.65	α-Al and θ-Al_2_Cu

**Table 4 materials-16-00436-t004:** The EDS analysis results for each point in Figure 8.

Spot	Al	Cu	C	B	Possible Material Phase Composition
1	97.86	1.80	0	0.34	α-Al
2	72.23	27.36	0	0.41	α-Al and θ-Al_2_Cu
3	0.51	0	30.56	68.53	B_4_C
4	68.55	31.25	0	0.20	α-Al and θ-Al_2_Cu

**Table 5 materials-16-00436-t005:** Tensile experiment results for aluminum alloys in different orientations of deposition.

Melting Rate (mm/min)	Direction	Tensile Strength/MPa	Elongation/%
120	Horizontal	179.60	5.89
130	Horizontal	200.31	6.94
140	Horizontal	230.56	7.46
150	Horizontal	228.25	6.91
140-B_4_C particle	Horizontal	285.84	10.24
120	Vertical	168.55	3.72
130	Vertical	193.36	3.39
140	Vertical	214.25	4.40
150	Vertical	219.23	4.02
140- B_4_C particle	Vertical	270.90	6.29

## Data Availability

Not applicable.

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
