# Peer review of "The Effect of B4C Powder on Properties of the WAAM 2319 Al Alloy"

_materials, 2023, doi:10.3390/ma16010436_

Round 1
Reviewer 1 Report
The paper is devoted to the study of the effect of B4C powder on the properties of the WAAM 2319 Al alloy. The reviewer has a number of comments and questions.
The paper can be published in the journal after a minor revision:
1. In the paper did not pay attention to the concentration of boron carbide? What amount was injected? Please indicate in the title of the paper and the abstract.
2. The use of such a composite can be used primarily for neutron capture. Please note the authors' attention to the following publication: 10.1016/j.jallcom.2021.160266
3. There is no data about the original boron carbide powder (particle size, morphology, XRD).
4. Does the formation of compounds such as aluminum boride or aluminum carbide occur in this cladding?
5. Figure 9, not enough images of the indenter, please integrate into figure 9
6. There is also a question about homogeneity. How well the mixture is mixed? An additional method FIB 10.1016/j.matchar.2018.08.044 is desirable
7. To what extent are intermetalides currently used, how resistant are your materials to oxidation in corrosive environments 10.3390/met12071089
8. Please also make additional references to the publisher's journal in the literature. The list of references is correct, it is commendable to see such an analysis
9. Figure 2, how much does cutting affect the properties of the sample? Or should it be neglected?
10. Figure 5, maybe we should specify the phase content by the Riveld method?
Author Response
Responses to the reviewer #1
Thanks very much for the detailed comments. We have benefited a lot from your advice. As for the shortcomings of the article, we have made corresponding modifications according to your review comments. As for your suggestions, we are also very recognized, and ready to adopt. Finally, thank you again for your comments.
- In the paper did not pay attention to the concentration of boron carbide? What amount was injected? Please indicate in the title of the paper and the abstract.
Authors’ Response: We are grateful for the suggestion. We are sincerely sorry for this problem and thank the referee for pointing those out. The study in this paper is mainly about the influence of adding B4C on the microstructure and properties of aluminum alloy, and the content of adding B4C in aluminum alloy is controlled by the injection rate. The powder injection rate at 25g/min.We have modified Table2.That is, the content of B4C in aluminum alloy in this paper is the same, so it is not further explained here. The effect of B4C content on microstructure and properties of aluminum alloy will be further studied.
- The use of such a composite can be used primarily for neutron capture. Please note the authors' attention to the following publication: 10.1016/.jallcom.2021.1602663.
Authors’ Response: Thank you for your advices. We will pay more attention to the neutron capture of this composite, and the article is useful for our next study.
- There is no data about the original boron carbide powder (particle size, morphology, XRD).
Authors’ Response: We are grateful for the suggestion. The data of boron carbide powder was described in line153-157. And the macroscopic morphology and energy spectrum analysis of B4C powder was shown in Fig.6
- Does the formation of compounds such as aluminum boride or aluminum carbide occur in this cladding?
Authors’ Response: We are grateful for the suggestion. The melting point of B4C is about 660 ℃. The maximum temperature of our added aluminum molten pool will not be higher than 2000 ℃. B4C, as a heterogeneous nucleating agent, will remain solid in the whole process of addition and will not react to form others.
- Figure 9, not enough images of the indenter, please integrate into figure 9.
Authors’ Response: Thank you for your careful review of our manuscript and your suggestion. we have made a change of it.
- There is also a question about homogeneity. How well the mixture is mixed? An additional method FIB 10.1016/i.matchar.2018.08.044is desirable.
Authors’ Response: We are grateful for the suggestion. We use powder feeder sent B4C powder to weld pool. The powder is blown out by the shield gas flow in the powder feeder,in this process, the mixture can be mixed enough before enter into the molten pool. Thank you for your advice of this article, we will absorb this good method in next study.
- To what extent are intermetalides currently used. how resistant are your materials to oxidationin corrosive environments 10.3390/met12071089.
Authors’ Response: We are grateful for the suggestion. It is also one of the key research directions to analyze the influence of corrosion resistance and wear resistance of aluminum alloy composite coatings with different B4C content. We will also consider this direction in the following research.
- Please also make additional references to the publisher's journal in the literature. The list of references is correct. it is commendable to see such an analysis.
Authors’ Response: We are grateful for the suggestion. We have modified literature, and cite an relevant article of publisher’s journal.
- Figure 2, how much does cutting affect the properties of the sample? Or should it be neglected?
Authors’ Response: We are grateful for the suggestion. Electric spark line cutting was used for cutting sample, and the sample was ground to remove the material property changes caused by cutting. Therefore, cutting will not affect the test of mechanical properties of the sample.
- Figure 5, maybe we should specify the phase content by the Riveld method?
Authors’ Response: We are grateful for the suggestion. This is an good comments. In this work, we just studied the effect of boron carbide powder on the microstructure and mechanical properties of 2319 aluminum alloy additive was studied. As you pointed out, it will be an important part of our next research.

Reviewer 2 Report
The article is well written. It presents interesting research on wire arc additive manufacturing process with the double-pulse melting electrode inert gas-shielded welding for 2319 aluminum alloy fabrication. The authors investigated the mechanisms of grain refinement and improvement of mechanical properties of the alloy when the B4C additive was used.
However, unfortunately, the results' innovation and novelty are not well indicated, although the work is a correct analysis of the posed research problem. Therefore, I strongly suggest emphasizing the innovativeness of your work in comparison to the achievements in this field already described in the literature, especially in the contexts of the section "1. Introduction" and the presented research articles concerning the investigated topic.
Also, the Abstract is not well written a lot of wordy sentences. Therefore, it is hard to read. Please read it carefully and try to paraphrase the sentences to make the content clear.
Minor remarks:
- line 78: addition of B4C: how much? Any quantification?
-line 116: uneven in what sense?
- lines 139 -149: what does it mean that peak width changed a little? You should, in my opinion, indicate how much? A little is not a measure. This change is not visible in Fig 5. Also, it would be interesting to calculate the grain size from the XRD data, as you can see the peak width change, and compare with the SEM results. There is also some inaccuracy in the Fig 5 caption and the text. Of the article. In line 142, you indicate Fig 5e, but there is no Fig 5e. Please check it carefully.
